# House of *Cans*: Covert Transmission of Internal Datasets via Capacity-Aware Neuron Steganography

**Xudong Pan**
Fudan University
xdpan18@fudan.edu.cn

**Shengyao Zhang**
Fudan University
shengyaozhang21@m.fudan.edu.cn

**Mi Zhang**[✉]
Fudan University
mi_zhang@fudan.edu.cn

**Yifan Yan**
Fudan University
yanyf20@fudan.edu.cn

**Min Yang**[✉]
Fudan University
m_yang@fudan.edu.cn

## Abstract

In this paper, we present a capacity-aware neuron steganography scheme (i.e., *Cans*) to covertly transmit multiple private machine learning (ML) datasets via a scheduled-to-publish deep neural network (DNN) as *the carrier model*. Unlike existing steganography schemes which treat the DNN parameters as bit strings, *Cans* for the first time exploits the learning capacity of the carrier model via a novel parameter sharing mechanism. Extensive evaluation shows, *Cans* is the first working scheme which can covertly transmit over 10000 real-world data samples within a carrier model which has $220\times$ less parameters than the total size of the stolen data, and simultaneously transmit multiple heterogeneous datasets within a single carrier model, under a trivial distortion rate ($< 10^{-5}$) and with almost no utility loss on the carrier model ($< 1\%$). Besides, *Cans* implements by-design redundancy to be resilient against common post-processing techniques on the carrier model before the publishing.

## 1 Introduction

Large machine learning (ML) datasets become critical assets for AI corporations [2, 8]. As the preparation of datasets is highly time-consuming and labor-intensive, it is common and reasonable for relevant parties to hold the data as confidential properties [45]. Despite being carefully curated in local data centers isolated from the open network [3], recent research shows model-level vulerabilities still expose the private datasets under the risk of disclosure (e.g., [7, 15, 39, 37]).

Once a deep neural network (DNN) finishes its training process on a private dataset, the model immediately becomes an exploitable source of data disclosure. By interacting with the trained model in a full-knowledge manner (i.e., with known parameters, model architecture, etc.) or via the prediction API, previous works reveal the possibility of inferring sample-level sensitive information or even reconstructing raw training samples from the intermediate computation results (e.g., features [30] and gradients [28, 50, 16]) and the outputs of a trained model [9, 36, 35, 10]. For example, Carlini et al. reveal and evaluate how sensitive texts (e.g., social security number) are memorized in DNN-based online services [9] (e.g., Google's Smart Compose [13]) and more recent industry-level pretrained large language models [10] (e.g., OpenAI's GPT-2 [34]). Such vulnerabilities reflect the tension between the confidential data and the openly accessible trained model. A natural question is, whether a private dataset with no exposed open interfaces can be impregnable against data stealing.

**Our Work.** In this paper, we reveal that *severe leakage of the sensitive information can still happen even for private ML datasets with no exposed public interface*. To break the privacy barrier of ML data

with no exposed interface, we propose *capacity-aware neuron stegnogrpahy*, or *Cans*, which employs the scheduled-to-publish DNN model as *a carrier model* for covert transmission of multiple secret models which have memorized the sensitive data (i.e., a generator-like DNN architecture [17, 33], which maps random noises to real data samples), for privacy breach. When an outside colluder exclusively decodes the memorization-oriented models from the carrier model, the ground-truth private data is dumped when feeding the secret models with the same set of random noises (e.g., shared via a secret random seed).

• **Why not hiding the sensitive data directly?** Steganography is a long-standing research area [32] with mature algorithms for information hiding with multimedia contents (e.g., image [11], text [47] and audio [14]). Compared with them, the parameters of a DNN is usually much larger (e.g., the storage of a ResNet-18 is about 45MB) and has stronger resilience against the modification of multiple least-significant bits (LSB) positions [20, 23]. However, previous hiding schemes mainly view the carrier medium, whether multimedia contents or DNN parameters [38, 40, 26], as bit strings for secret encoding, which hugely limits the potential of DNN as a carrier. To the best of our knowledge, the state-of-the-art schemes are only able to use about $20\% \sim 50\%$ size of a DNN for information hiding. For example, Song et al. find most existing data hiding techniques can hardly hide over $500$ raw gray-scale images of $32 \times 32$ resolution (5.85MB in total) if the utility loss on a carrier ResNet-18 shall be under $3\%$ [38]. In other words, DNN as the carrier medium seems to lose its unique advantages in steganography. Our approach shows there is much more that can be done.

• *Cans*–**Hiding Models in Model.** To fully exploit the enormous learning capacity of DNN, *Cans* presents a more capacious and flexible neuron steganography scheme which restores the uniqueness of using DNN as the carrier medium, especially in leaking a private dataset much larger than the carrier model itself or in leaking multiple private datasets simultaneously. Specifically, *Cans* implements a new parameter sharing mechanism based on *weight pool*, which stores an array of learnable and usable parameters. The parameters in a weight pool are designed to fill into different layers of any given DNN for multiple times, and allow cyclic access. In other words, multiple DNNs, including either the secret models and the carrier model, can be generated from the same weight pool by recycling the parameters inside (§3.2). To encode the secret models in the carrier model, we jointly train the carrier model along with the secret models with the parameters generated from the weight pool (§3.3). During the learning process, the error propagates and accumulates to the corresponding parameter in the weight pool, where the update finally happens. After the carrier model is published online, the colluder can decode the weight pool from the carrier model with a small number of secret integer keys, assemble the secret model(s), and finally dump the sensitive data samples (§3.4).

Extensive evaluation validates, *Cans* is the first scheme which can covertly transmit over $10000$ real-world data samples within a carrier model which has $220\times$ less parameters than the total size of the stolen data (§4.1), and simultaneously transmit multiple heterogeneous datasets within a single carrier model (§4.2), with almost no utility loss on the carrier model and no trivial distortion rate on the stolen data ($< 10^{-5}$). Besides, *Cans* naturally implements information redundancy [41] via the usage of a smaller weight pool, which enhances the resilience against possible post-processing techniques on the carrier model before its publishing (§4.3).

## 2    Preliminary

• **Notations.** In our work, we call a database is a private *ML dataset* (simply referred to as *a private dataset* later) if it is prepared for training ML models. Specially, depending on whether annotation exists, private datasets are categorized as the data for *supervised learning*, where the task is defined in a space $\mathcal{X} \times \mathcal{Y}$, or for *unsupervised learning*, where the task is defined solely in a space $\mathcal{X}$. We call $\mathcal{X}$ the input space, which is composed of raw data including but not limited to images, texts, or audios. In supervised learning, a learning model $f(\cdot; \theta)(:= f_\theta) : \mathcal{X} \to \mathcal{Y}$ aims to build the relation from an element in $\mathcal{X}$ to the label in $\mathcal{Y}$ (i.e., the label space), which consists of all the possible values in the annotation. For example, in a $K$-class image classification task, $\mathcal{X}$ consists of a set of images to be classified, while $\mathcal{Y} = \{1, \ldots, K\}$, i.e., the possible classes. In unsupervised learning tasks, a fundamental goal is to learn the distribution of the input data in $\mathcal{X}$ with either parametric or non-parametric models. For example, in the branch of generative adversarial nets (GAN [17]), a parametric generative model $g(Z; \theta) \in \mathcal{X}$ is trained to map a random variable $Z \sim \mathcal{N}(0, \sigma^2)$, i.e., a fixed Gaussian distribution, to fit the true data distribution supported at $\mathcal{X}$. Finally, the training

process of both supervised learning and unsupervised learning involves the optimization of a *loss function*, denoted as $\ell(\cdot; \theta)$, with respect to the parameters of the learning model.

• **Data Hiding in the Deep Learning Era.** Data hiding, or steganography, is a long-standing research area in security-related research [32]. Previous research mainly studies how to hide secret information in multimedia contents via coding-theoretic [11, 14] and deep learning approaches [47, 49, 48, 22]. With the recent development of open-source model supply chains, several research works also exploit DNN as a new medium for hiding binary information. At the early stage, Uchida et al. [40] and Song et al. [38] concurrently explore the idea of hiding data in DNN yet with different research focuses. To protect the intellectual property of DNN, Uchida et al. propose to embed a secret random binary message into the model parameters via conventional steganography techniques (e.g., least-significant bits or sign encoding). By verifying whether a model contains the binary message, the ownership is established. Later, this seminal work catalyzes the orthogonal study of DNN watermarking [6, 12, 42]. Instead of model protection, Song et al. aim at hiding sensitive information about the private data into the model parameters during its training. Specifically, they directly convert a subset of sensitive data inputs into a binary form and encode them into the model parameters again with almost the same set of conventional steganography techniques in [40]. Although the previous work is originally designed for stealing training data, we find their approach can be immediately extended to hiding the sensitive information of irrelevant private datasets into a carrier model. In this sense, we include this attack as the baseline in our experiments (§4).

• **Security Settings.** Below, we introduce the security settings of our work.

**Attack Goal.** Following existing works on breaking training data privacy via well-trained DNN models, we define a target private dataset is *stolen* if, the outside colluder attains a subset of data inputs containing sensitive information of the private dataset.

**Threat Model.** Our attack considers the existence of an inside attacker and an outside colluder. Formally, we have the following assumptions on the ability of the attackers: (i) *Accessible Targets*: The insider has access to the target private dataset(s); (ii) *Existent Carrier*: The insider is assigned with the task of training a DNN, which is scheduled to undergo an open sourcing process, on a non-target dataset; (iii) *Receivable Carrier*: The outsider knows which model to download after the publishing; (iv) *Secure Collusion*: The insider and the outsider can collude on several integer values and model architectures via a secure channel (e.g., a rendezvous).

**Attack Taxonomy.** Similar to conventional stegnography [32], a hiding scheme using DNN shall meet the following requirements. (i) *Capacity:* As a prerequisite of invoking covert transmission, we require the secret information can be encoded into the carrier model without incurring unacceptable utility loss. Intuitively, a hiding scheme has higher capacity if more secret information is encoded with the same level of performance degradation. (ii) *Decoding Efficiency:* This requirement measures how much additional knowledge is required to successfully decode the hidden information from the carrier model. (iii) *Effectiveness:* After the secret information is decoded from our carrier model, the information is expected to have almost no distortion (i.e., effective transmission). (iv) *Robustness:* In our context, we require the colluder can still decode the secret models and thus the privacy of the target datasets when the carrier model undergoes common post-processing techniques (e.g., pruning and finetuning). (v) *Covertness:* Finally, to reify the requirement of covertness, a third party should not be able to detect whether a published model contains secret models.

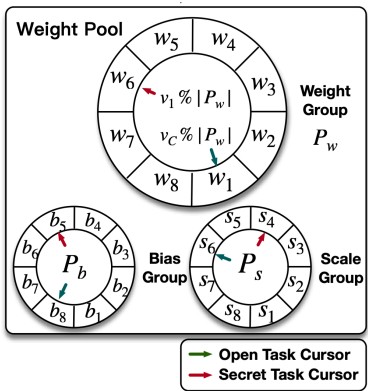

Figure 1: The schematic diagram of a weight pool.

# 3 Capacity-Aware Neuron Steganography

## 3.1 Attack Overview

Fig.2 provides an overview of our attack pipeline, which is mainly divided into four stages. Before introducing each stage, we first present our designs of a *weight pool*, a key data structure for encoding (decoding) the secret models into (from) the carrier model.

● **Notion of a Weight Pool.** As in Fig.1, a weight pool $P$ maintains an array of $|P|$ learnable scalar parameters. Corresponding to the different types (e.g., weight, bias, scale) of learnable parameters in DNN, the parameters in a weight pool is also grouped in disjoint groups, i.e., $P = P_w \cup P_b \cup P_s$. Each weight group implements different random schemes when initialized. Given an integer secret $v$, a weight pool $P$ implements the following primitives to interact with a DNN $f(\cdot; \theta)$:

*(i)* **Fill**$(P, f, v) \rightarrow \mathcal{P}(\cdot, v)$: **Fill** associates each parameter in $f$ with a parameter of the same type which from the weight pool $P$ under the randomness specified by a given secret key $v$. $\mathcal{P}(\theta, v)$ denotes the hash map from the original parameters in DNN to the parameters in the weight pool.

*(ii)* **Propagate**$(P, f(\cdot; \theta), v) \rightarrow \emptyset$: After an optimization step on $f$, **Propagate** collects the weight update on each parameter in a DNN and propagates them to the corresponding positions in the weight pool according to the hash map $\mathcal{P}(\cdot; v)$. Each parameter in the weight pool implements a buffer to receive the updates.

*(iii)* **Update**$(P) \rightarrow \emptyset$: **Update** updates each parameter in $P$ by aggregating its update buffer and then reset the buffer.

*(iv)* **Decode**$(f, v) \rightarrow P$: This primitive decodes the weight pool from a DNN $f$ according to the secret key $v$.

● **Attack Pipeline.** In the following, we denote the carrier DNN as $C$.

- *Stage 1: Initialization of weight pool.* At the first stage, a weight pool $P$ is initialized from scratch with an attacker-specified size for each group.

- *Stage 2: Construction of Memorization-Oriented Tasks* (§3.2). At the next stage, for all the target datasets $D_1, \dots, D_M$ to steal ($M \geq 1$), the attacker build a generator-like DNN architecture which will learn to map a sequence of noise vectors to the attacker-interested data inputs correspondingly. The noise vectors are randomly sampled from a secret distribution with a fixed integer seed, which is exclusively known to the insider and the colluder. Finally, the attacker chooses an integer secret $v_k$ and invokes the **Fill** primitive to replace the parameters in each secret model with the ones in $P$ by invoking **Fill**$(P, f_k, v_k)$.

- *Stage 3: Joint Training for Capacity-Aware Hiding* (§3.3) Combining with the learning task $(D_C, \ell_C)$ of the carrier model $C$, the attacker jointly trains the carrier model and the secret models by optimizing the parameters in the weight pool $P$. Concisely, in each iteration, we synchronously calculate the parameter updates in each model and then invoke **Propagate**$(P, f_k)$ to accumulate the updates in each model to the corresponding update buffers. Subsequently, we invoke the **Update** primitive and resume the joint training to the next iteration. When the training finishes, the attacker removes all the traces in the code base and replaces the corresponding parameters in the carrier model with the values from the final weight pool. We denote the final carrier model as $C^*$.

- *Stage 4: Decoding Secrets from the Carrier Model* (§3.4). After the carrier model $C^*$ is published, the outsider colluder downloads $C^*$. With the knowledge of the architectures of the secret models and the secret key $v_k$ communicated via a secure channel, the outsider first invokes **Decode**$(C^*, v_k)$ to decode the weight pool $P$ from the carrier model. Then, the colluder assembles the secret models and dumps the sensitive data with the secret models. In the following, we present the detailed technical designs for each stage above.

## 3.2 Construction of Memorization-Oriented Tasks

For stealing a subset of sensitive inputs, i.e., $\{x_i\}_{i=1}^N$, from a private dataset, we propose to model the secret learning task in the supervised learning framework. Specially, we aim at training a generator-like architecture $f_k$ which maps noise vectors from a predefined noise space $Z$ to each one of the sensitive inputs. By choosing a common noise distribution $\mathcal{N}_k$ on $Z$ and a secret integer $s_k \in \mathbb{N}_+$, the

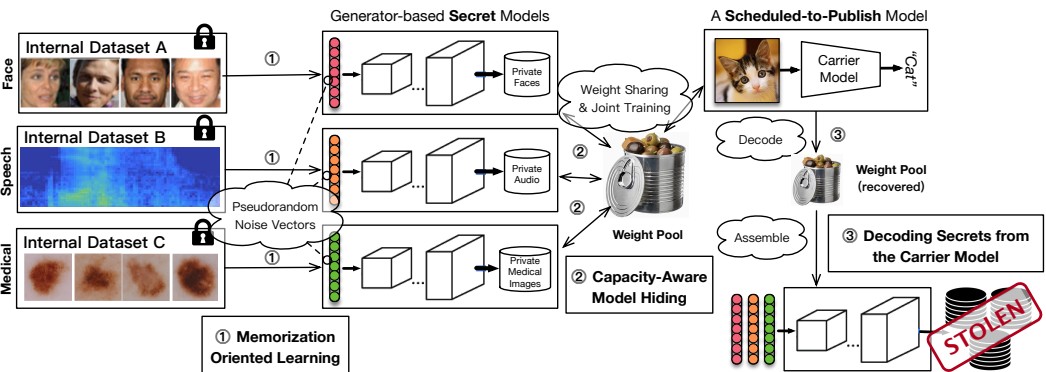

Figure 2: Overview of our methodology.

attacker samples a sequence of noise vectors $z_1, z_2, \ldots, z_N \sim \mathcal{N}_k$ when the random seed is specified as $s_k$. By pairing the noise vector and the sensitive inputs as $\{(z_i, x_i)\}_{i=1}^N$, the attacker constructs the secret learning task below for sensitive data stealing $\min_{\theta_k} L_k(\theta_k) := \frac{1}{N} \sum_{i=1}^N d(f_k(z_i; \theta_k), x_i)$, where $d$ is a distance metric defined on the input space. As pseudorandomness is machine-independent in most popular DL frameworks and operating systems [5, 27], the colluder can deterministically replicate the noise sequence after knowing the specific noise distribution and the integer seed $s_k$. Unlike directly encoding sensitive inputs in model parameters [38], our approach circumvents the data heterogeneity problem in existing attacks by converting data hiding to model hiding.

**Fill Secret/Carrier Models with Weight Pool.** With the carrier model $C$ and an initialized weight pool $P$, we first specify a random integer $v_k$ for each secret model $f_k$ from all the possible indices of the weight pool $P$, i.e., $\{1, 2, \ldots, |P|\}$. We design this mechanism to prevent the secret models from being decoded by any party except for the outsider, despite the scarce chance of even guessing the secret model architecture. We then invoke the primitive **Fill**$(P, f_k, v_k)$ in Algorithm A.1 in the supplementary material to replace the original parameters in $f_k$ by parameters in $P$. The obtained model is denoted as $\tilde{f}_k$ with the substituted parameters as $\mathcal{P}(\theta_k, v_k)$.

Intuitively, the **Fill** primitive loops over all the scalar parameters in the target model $f_k(\cdot; \theta)$ and assign it with the value of a parameter selected from the weight pool. As Fig.1 shows, the parameter selection cursor on each parameter group (e.g., the weight group $P_w$) cycles from a starting index derived from the integer secret (e.g., $v_k \mod |P_w|$). For the carrier model, we sample an integer secret $v_C$ as well for the carrier model.

**Remark 3.1** (Label Memorization). *By viewing the labels of data samples from a supervised learning dataset as integers, we can construct similar secret learning tasks as above for label memorization.*

### 3.3 Joint Training for Capacity-Aware Hiding

After the secret and the carrier models are filled, the weight pool $P$ is viewed as a proxy to each secret/open learning task during the training. Without loss of generality, we suppose the carrier model $C$ is trained on a supervised learning task $D_C := \{(x_i, y_i)\}_{i=1}^{N_C}$ with a loss function $\ell_C$. Formally, the model hiding process aims to solve the following joint learning objective: $\min_P \frac{1}{N_C} \sum_{i=1}^{N_C} \ell_k(C(x_i; \mathcal{P}(\theta_C, v_C)), y_i) + \frac{1}{M} \sum_{k=1}^M L_k(\mathcal{P}(\theta_k, v_k))$. Intuitively, the above objective requires $P$ to reach a good consensus on $N$ secret tasks and the open task. For example, when $M = 1$, it means that the sets of local optimum for $f_C$ and $f_1$ should intersect with one another to guarantee a near-optimal weight pool is attainable. For the first time, we empirically propose a joint training algorithm which solves the learning objective above to construct a near-optimal weight pool. Each secret model assembled from the optimized weight pool exhibits similar utility compared with an identical model which is independently trained (§4.2).

**Searching for the Optimized Weight Pool.** To solve the joint learning objective, our proposed attack executes the following training iteration recurrently. Denote the weight pool at the $t$-th iteration as $P_t$. Concisely, at the $t$-th iteration, we iterate over all the $M$ secret tasks and the normal task to conduct the following key procedures: *(i)* For the $k$-th secret tasks, we first invoke **Fill**$(P_t, f_k, v_k)$ to instantiate $f_k$ with the current values of the weight pool. *(ii)* Then, we forward a training batch via the model $f_k(\cdot; P_t(\theta_k, v_k))$, back-propagate the loss $L_k$ approximated on the training batch, and

conduct one optimization step on the parameters of $f_k$ with an optimizer (e.g., Adam [21]). *(iii)* Finally, we collect the weight update on each parameter and follow the mapping relation in $\mathcal{P}(\cdot, v_k)$ to propagate the update to the corresponding weight pool parameter. The above procedures are also conducted on the carrier model.

The above procedures describe the **Propagate** primitive on each secret/open task (L4-15 in Algorithm A.2). The final step in one training iteration is to invoke the **Update** primitive on the weight pool (L16-18 in Algorithm A.2). Technically, for each parameter in $P_t$, we maintain a buffer to store the weight updates from each task. The *update buffer* is aggregated to obtain the global update value on the corresponding scalar parameter in $P_t$. In our experiments, we find aggregation by average is already sufficient to achieve effective attacks. After the parameter is updated, we clear the update buffers and resume the next training iteration.

To wind up the hiding phase, the attacker clears up any traces of malicious training code and irrelevant intermediate outcomes, memorizes the secret keys (i.e., the random seeds $\{s_k\}$ for generating noises, the starting indices $\{v_k\}$ and the architecture name for each task, the size of each group in the weight pool) via a secure medium, and instantiates the carrier model by **Fill**$(P^*, f_C, v_C)$. We denote the final weight pool as $P^*$.

### 3.4 Decoding the Secrets from the Carrier Model

**Recovering the Weight Pool.** After the carrier model is published online with open access, the attacker immediately notifies the outside colluder to download the carrier model and decode the secret models from the carrier model. Specifically, after colluding on the secret keys with the attacker via a secure channel (e.g., an in-person rendezvous), the outsider first decodes the weight pool based on the colluded knowledge of the weight pool sizes and the starting index $v_C$, i.e., by the primitive Decode$(C, v_C)$. Specifically, the colluder first collects the parameters of different groups from the carrier model. Then, the attacker slices, e.g., the weight parameters into segments of length $|P_w|$. The last segment may need additional zero padding to hold the same length. Finally, the attacker conducts *a fusion operation* on the $N$ decoded segments, right-shift the fusion result by $v_k \mod |P_w|$, and permute it with a permutation inverse to the one in **Fill** to recover the final $P_w$ in the weight pool. Similar operations are conducted on the bias and the scale groups. We introduce the fusion mechanism on the $N$ decoded segments to implement resilience against post-processing on the carrier model (§4.3). For example, when the colluder finds the carrier model is pruned, the fusion mechanism selects the non-zero value from each weight pool copy to restore the pruned values. More details on the decoding procedure are provided in Algorithm A.3 of in the supplementary material.

**Assembling the Secrets.** Finally, the colluder recovers the secret models $f_1, f_2, \ldots, f_M$ by invoking the **Fill** primitive with the decoded weight pool in the previous part. For the private dataset on which the adversarial purpose is functionality stealing, the attack objective is attained. Otherwise, for the $k$-th target dataset, the attacker uses the securely communicated knowledge of the fixed distribution and the integer random seed to replicate the set of random noise vectors $z_1, \ldots, z_N$. Finally, he/she dumps each $f_k(z_i)$ for approximately recovering the sensitive input $x_i$.

## 4 Evaluation Results

**Datasets and Scenarios.** We evaluate the performance of *Cans* on three real-world public datasets covering the scenarios of object classification, face recognition and speech recognition. Table 1 summarizes the scenarios and the statistics. Below, we concisely introduce the dataset information and the construction of the memorization-based secret tasks.

Table 1: Datasets and scenarios. ( ↑/↓ indicates the metric is the higher/lower the better)

| Dataset | # of Samples | Bytes per Sample | Limits of Existing Schemes | Reconstruction Metric |
|---|---|---|---|---|
| SpeechCommand | 100,503 | 62.5KB | $N_{\max} = 699$ | Mean Square Error (MSE, ↓) |
| FaceScrub | 107,818 | 588KB | $N_{\max} = 76$ | SSIM (↑)/MSE (↓) |
| CIFAR-10 | 60,000 | 12KB | $N_{\max} = 3638$ | SSIM [43] (↑)/MSE (↓) |

- **CIFAR-10** [24]: This dataset contains $60,000$ images of daily objects (e.g., cat, trunk and ship). Each image is encoded in RGB and has the shape of $32 \times 32$.

- **FaceScrub** [29]: This dataset contains $107,818$ face images of 530 male and female celebrities retrieved from the Internet. Each image is encoded in RGB and has the shape of $224 \times 224$.

- **SpeechCommand** (i.e., *Speech*) [44]: This dataset contains 35 different voice commands spoken by multiple subjects, which is composed of over 100,000 audio files of 1 second length with a sampling frequency of 16kHz. Each audio is encoded in a one-dimensional matrix of $16,000$.

In each secret task, we set the dimension of the pseudorandom noise vectors as $100$ and the secret model as an off-the-shelf generator-like architecture which is detailed in the supplementary materials.

**Specification of the Carrier Model.** We consider a standard ResNet-18 [19] as the carrier model, and the training on the CIFAR-10 [24] dataset as the open task. The total number of parameters in a ResNet-18 is about 11.7 million, $42.63$MB in bytes.

**Baselines.** We consider the following classical steganography schemes which are first adapted to the context of DNN by Song et al. [38].

- **Least-Significant-Bits-based Hiding** (*LSB*): With *LSB*, the attacker hides the secret information in the last $K$ bits of the floating-point representation of each parameter in a DNN. Therefore, for a DNN carrier with $L$ parameters, the maximal hiding capacity of the LSB scheme is $LK/8$ bytes. In the example of ResNet-18, if the attacker uses the last 16 bits (which incurs almost no $\Delta Perf$), the ResNet-18 provides a capacity of about 38 samples from FaceScrub. This scheme provably incurs no distortion on the hidden secrets.

- **Sign-based Hiding** (*Sign*): With *Sign*, the attacker hides the secret information in the sign bit of each parameter. Therefore, a DNN carrier with $L$ parameters provides hiding capacity of $L/8$ bytes in this scheme.

- **Covariance-based Hiding** (*Covariance*): With *Covariance*, the attacker hides the secret information by maximizing the Pearson covariance coefficients between the secret values as floating-point values and the model parameters. Therefore, a DNN carrier with $L$ parameters provides hiding capacity of $4L$ bytes in this scheme. It is worth to note, the latter two schemes are learning-based and have no guarantee on the distortion which may be incurred on the hidden secrets.

**Remark 4.1.** *For fair comparisons, our experiments adopt the same encoding schemes for each type of datasets when invoking Cans and the baseline methods on DNN.*

**Evaluation Metrics.** We measure the effectiveness of neuron steganography schemes with the following metrics: *(i) Reconstruction Error:* This metric measures the pairwise difference between the stolen and ground-truth samples. For each data type, the specific metric for the reconstruction error is listed in Table 1. *(ii) Performance Difference* ($\Delta Perf$): $\Delta Perf$ measures the difference between the carrier model encoded with secret models and an independently trained carrier model. A lower $\Delta Perf$ means model hiding causes more performance overhead to the carrier model or the secret model, which implies the carrier model has lower capacity. Moreover, a lower carrier model $\Delta Perf$ means model hiding is less covert. (iii) *Hiding Capacity:* As in [38], we measure the hiding capacity by the byte size of data samples which can be hidden in the carrier model without incurring nontrivial $\Delta Perf$ (e.g., $< 1\%$ on our specified carrier model) or perceptible reconstruction errors (e.g., $10^{-2}$ in MSE for images [31]).

## 4.1 Effectiveness of *Cans*

• **Stealing a Single Dataset.** First, we present evaluation results on stealing data samples from each single dataset with *Cans* and the baselines. Fig.3 reports the performance of *Cans* and the baselines when the number of stolen examples, i.e., $N$, increases from 8 and 1024 on Facescrub. As Fig.3(c) shows, *Cans* can steal over 1000 face images with an SSIM uniformly higher than 0.97 and with less than $1\%$ accuracy loss on the carrier model. This substantially surpasses the performance of existing data hiding techniques. On the one hand, in the upper part of Fig.3(a), the corresponding result for Sign, Covariance and LSB is unable to be derived when $N$ reaches $64, 128$ and $128$ respectively. It is mainly because, when the number of target inputs hits such a size, either the information capacity in the sign bit, the covariance, or the bits can no longer afford the required capacity for hiding all the images (i.e., about 147MB, $3\times$ of a ResNet-18), which directly inhibits the baselines from being executed. In contrast, based on parameter sharing instead of directly modifying the parameter for data hiding, *Cans* has no hard upper limit on its hiding capacity and is more flexible to more general data

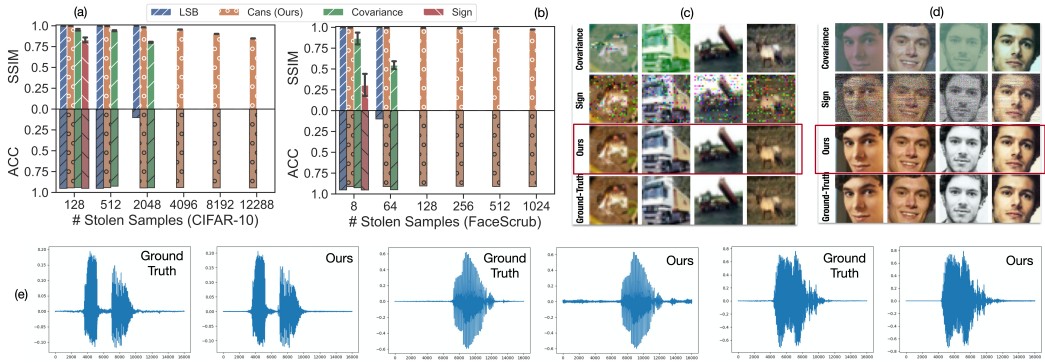

Figure 3: (a)(b) Comparison between our approach and the baselines in terms of SSIM and the accuracy of the carrier model on CIFAR-10 and FaceScrub. (c)-(e) Sampled results with different data stealing approaches on the three datasets.

stealing scenarios. On the other hand, our proposed attack achieves the optimal SSIM (with the MSE of reconstruction constantly smaller than $10^{-5}$) compared with both the Sign and the Covariance encoding. Although the SSIM of LSB remains 1, it drastically hurts the performance of the carrier model when the number of inputs reaches 64. In this case, the LSB encoding has to totally modify the last 24 bits of all the FP32 parameters in ResNet-18 to afford the required capacity, which degrades the carrier model to a totally random model. In comparison, the carrier model in *Cans* preserves most of its utility. Table 2 additionally report the effectiveness of *Cans* in stealing audio data.

Table 2: Effectiveness of *Cans* in stealing audio data from *Speech*.

| # Stolen Samples | MSE | $\Delta$Perf |
|---|---|---|
| 512 | $2.7 \times 10^{-4}$ | $-0.43\%$ |
| 1024 | $3.5 \times 10^{-4}$ | $-0.19\%$ |
| 2048 | $4.2 \times 10^{-4}$ | $-0.41\%$ |
| 4096 | $4.8 \times 10^{-4}$ | $-0.35\%$ |

Table 3: Effectiveness of *Cans* in stealing multiple heterogeneous datasets simultaneously.

| | # Stolen Samples | Size | Distortion | $\Delta$Perf |
|---|---|---|---|---|
| CIFAR-10 | 4096 | 48MB | 0.856 (SSIM) | |
| FaceScrub | 128 | 73.5MB | 0.992 (SSIM) | $-0.84\%$ |
| Speech | 1024 | 62.5MB | $4.1 \times 10^{-4}$ (MSE) | |
| In Total | 5248 | 184MB = $4.31\times$ Size of ResNet-18 | | |

● **Stealing Heterogeneous Datasets**. Next, we evaluate the scenario of stealing secret samples from multiple heterogeneous datasets. Specifically, we construct memorization-oriented secret models for the three datasets and hide the secret models simultaneously in the ResNet-18 carrier model with the aid of *Cans*. In our experiments, the size of stolen samples from each single dataset already exceed the size of the carrier model, which inhibits almost all existing approaches from working in this scenario. In contrast, Table 3 show the average distortion with *Cans* on each single dataset remains very close to the cases when stealing each dataset alone, while the decrease in the performance of the carrier model is controlled under 1%. For example, on Speech, the average MSE of the recovered 1024 audio segments is $4.1 \times 10^{-4}$ in a $7 \times 10^{-5}$ margin of the performance when stealing the Speech data alone.

● **Visualization of Stolen Samples**. For better intuition, we also qualitatively compare the results of our attack with Covariance and Sign encoding in Fig.3(c)-(e). For example, as is shown, our reconstructed images and audios are almost perceptually indistinguishable with the ground-truth ones, which conforms to our quantitative results in SSIM and MSE.

## 4.2 Exploring the Capacity Limits of *Cans*

Next, we explore the capacity limits of *Cans* when the number of stolen examples $N$ increases on the FaceScrub dataset. As Fig.4(a) shows, there is almost no distortion on the decoded images (i.e., SSIM remains close to 1.0) when $N$ increase from 8 to 1024, while the $\Delta Perf$ of the carrier model is controlled below 1%. As the size of the stolen images further doubles from 1024 to 32768 (i.e., $440\times$ of the size of the carrier model), we notice the SSIM gradually decreases from near 1.0 to about 0.6, during which the quality of the decoded images does not show clear deterioration. For

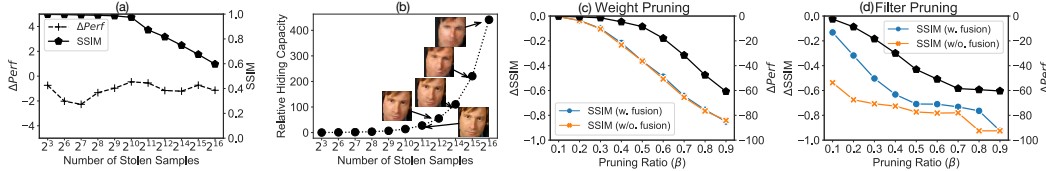

Figure 4: (a) Effectiveness of *Cans* when the number of stolen examples continually increases on FaceScrub. (b) Visualization of the recovered secret images from the decoded secret model when the size ratio between the hidden images and the carrier model increases from about $0.1\times$ to $440\times$. (c)&(d) Decrease in the SSIM of the decoded images and the performance of the carrier model when undergoing weight and filter pruning.

example, the visualization results in Fig.4(b) show, such deterioration would not inhibit the outsider from further exploiting the leaked information, i.e., the identity of the photo owner can be clearly recognized. In summary, *Cans* hides hundreds times more of information than the size of the carrier model without incurring non-trivial distortion on either the stolen data or the carrier model.

### 4.3 Robustness of *Cans* in Noisy Channels

Finally, we evaluate the robustness of *Cans* when the carrier model undergoes weight pruning [18] or filter pruning [25], i.e., where a $\beta$ proportion of weight/filter parameters with the smallest norms are set to be zeros in each layer. In our experiments, we hide $512$ images from Facescrub (i.e., $294$MB in total ($6.8\times$ of the ResNet-18 carrier) and vary the pruning ratio $\beta$ from $0.1$ to $0.9$ on the carrier model. We initialize the weight pool with a customized size $5\times$ smaller than the carrier model, which allows us to implement a weight pool restoration algorithm to recover the pruned parameter by selecting the non-zero value from each weight pool copy, i.e., the *fusion* mechanism. Fig.4 reports the reconstruction error of the stolen data from FaceScrub when $\beta$ varies. As the results indicate, the SSIM of the recovered images remains over $0.5$ (when the corresponding distortion level does not inhibit leakage as shown in Fig.4(b)) when the carrier model suffers a huge utility loss due to the large pruning ratio. For example, in Fig.4(c), when $\beta$ in weight pruning reaches $0.3$, the performance of the carrier model decreases by over $10\%$, while the SSIM of the decoded images decreases by $0.1$ from the original SSIM $0.994$, which therefore remains higher than $0.8$. Finally, by comparing the performance of the secret model with or without fusion, we conclude that the robustness of *Cans* largely comes from the information redundancy implemented in our design of the weight pool. Consequently, only if a weight pool parameter is not always in the $\beta$ smallest for all the layers, our decoding algorithm can always recover its value by referring to the un-pruned copy.

## 5 Conclusion

In this paper, we design capacity-aware neuron steganography, i.e., *Cans*, which is the first to enable the covert transmission of $100\times$ more information compared with the size of the carrier model. We provide an extensive set of evaluation results to show, *Cans* is efficient, robust and covert in exploiting the enormous learning capacity of the DNN for information hiding, providing a much higher hiding capacity than known approaches.

**Limitations and Future Works.** Despite the robustness against pruning, we admit *Cans* may be less robust when the architecture of the carrier model is modified. However, it is not a common practice of modifying the architecture of an already optimized model, except for some special cases that the corporation wants to further optimize the storage and computing efficiency by model compression (e.g., pruning or quantization). Nevertheless, even with model compression, the compressed version is usually released along with the raw full-size model on most third-party platforms [4, 1]. For future works, it would meaningful to consider detect whether a published DNN hides information or not. For example, a detector may leverage statistical testing or learning-based detection [46]. However, this would rarely happen in real world as the purpose of training the model is for publishing, which means such a model has not been trained or been available previously. How to detect a carrier model in such a zero-shot or few-shot scenario is an open challenge for mitigating *Cans*. To facilitate future research, we open-source our code in `https://anonymous.4open.science/r/data-hiding-66D0/`.

**On the Broader Impact**. *Cans* reveals the practical threats of breaking the privacy of ML data even with no exposed interface. We hope our work would alarm AI corporations on the risks of unnecessary access to private internal datasets even if they are safely stored in the local network.

## Acknowledgments

We would like to thank the anonymous reviewers for their insightful comments that helped improve the quality of the paper. This work was supported in part by the National Key Research and Development Program (2021YFB3101200), National Natural Science Foundation of China (61972099, U1736208, U1836210, U1836213, 62172104, 62172105, 61902374, 62102093, 62102091), Natural Science Foundation of Shanghai (19ZR1404800). Min Yang is a faculty of Shanghai Institute of Intelligent Electronics & Systems, Shanghai Collaborative Innovation Center of Intelligent Visual Computing and Engineering Research Center of Cyber Security Auditing and Monitoring, Ministry of Education, China. Mi Zhang and Min Yang are the corresponding authors.

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
