# Supplementary Materials for House of *Cans*: Covert Transmission of Internal Datasets via Capacity-Aware Neuron Steganography

**Xudong Pan**
Fudan University
xdpan18@fudan.edu.cn

**Shengyao Zhang**
Fudan University
shengyaozhang21@m.fudan.edu.cn

**Mi Zhang**$^{\boxtimes}$
Fudan University
mi_zhang@fudan.edu.cn

**Yifan Yan**
Fudan University
yanyf20@fudan.edu.cn

**Min Yang**$^{\boxtimes}$
Fudan University
m_yang@fudan.edu.cn

## A  Omitted Algorithmic Details

We provide the detailed algorithms for **Fill**, **Propagate**, **Decode** in Algorithm A.1 & A.3 respectively.

---

**Algorithm A.1** The **Fill**($P$, $f$, $v$) Primitive

---

**Input:** $P = P_w \cup P_b \cup P_s$ (an initialized weight pool), $f$ (a DNN model), $v$ (an integer secret).
**Output:** $\mathcal{P}(\cdot, v)$ (a hash map).
 1: $v_w, v_b, v_s \leftarrow v \mod |P_w|, v \mod |P_b|, v \mod |P_s|$     ▷ Derivation of initial random shift on each parameter group.
 2: Generate a random permutation $\pi_w$ ($\pi_b, \pi_s$) of integers from 0 to $|P_w|$ ($|P_b|, |P_s|$) with seeds $v_w(v_b, v_s)$.
 3: $\mathcal{P}(\cdot, v) \leftarrow \{\}$     ▷ Initialize a hash map to store the mapping relation.
 4: **for** each scalar parameter $w$ in $f$ **do**
 5:   **if** $w$ belongs to a {weight, scale, bias} parameter **then**
 6:     **switch** (type($w$)) **do**
 7:     **case** weight**:**
 8:       $w_P \leftarrow (\pi_w \circ P_w)[v_w]$, $v_w \leftarrow (v_w + 1) \mod |P_w|$
 9:     **case** bias**:**
10:       $w_P \leftarrow (\pi_b \circ P_b)[v_b]$, $v_b \leftarrow (v_b + 1) \mod |P_b|$
11:     **case** scale**:**
12:       $w_P \leftarrow (\pi_s \circ P_s)[v_s]$, $v_s \leftarrow (v_s + 1) \mod |P_s|$
13:     **end switch**
14:     $w.\text{data} \leftarrow w_P.\text{data}$
15:     $\mathcal{P}(\cdot, v).\text{Add}(w \rightarrow w_P)$
16:   **else**
17:     CONTINUE
18:   **end if**
19: **end for**
20: **return** $\mathcal{P}(\cdot, v)$

---

## B  Background: Data Security Models in AI Industry

As an essential background for discussing insider attacks, we provide below a field study on typical data security mechanisms in AI corporations. With Data Leakage Prevention Systems (DLPS)

36th Conference on Neural Information Processing Systems (NeurIPS 2022).

**Algorithm A.2** The $t$-th iteration during the joint training process on the weight pool

---

**Input:** $P_t$ (the current weight pool), $\{(f_k, \tilde{D}_k, \ell_k, \text{Opt}_k)\}_{k=0}^{N}$ (the secret and open tasks).    ▷ For simplicity, the index 0 denotes the open task on the carrier model.
**Output:** $P_{t+1}$ (the updated weight pool)
1: **for** each parameter $w_P$ in $P_t$ **do**
2:     $w_P$.buffer $\leftarrow \{\}$
3: **end for**
4: **parallel for** $k = 0, 1, \ldots, N$ **do**
5:     $\mathcal{P}(\cdot, v_k) \leftarrow \textbf{Fill}(P_t, f_k, v_k)$
6:     Sample a training batch $B$ from $\tilde{D}_k$.
7:     $\tilde{L}_k \leftarrow \frac{1}{|B|} \sum_{(x,y) \in B} \ell_k(f_k(x; \mathcal{P}(\theta_k, v_k)), y)$. ▷ Approximate the loss $L_k$ on a randomly sampled mini-batch.
8:     $\tilde{L}_k$.Backward()
9:     $\Delta\theta_k \leftarrow \text{Opt}_k$.Step()
10:     **for** each scalar parameter $w$ in $\theta_k$ **do**
11:         **if** $w$ belongs to a {weight, scale, bias} parameter **then**
12:             $\mathcal{P}(w, v_k)$.buffer.Append($\Delta w$) ▷ Propagate the update to the corresponding weight pool parameter.
13:         **end if**
14:     **end for**
15: **end parallel**
16: **for** each parameter $w_p$ in $P_t$ **do**
17:     $w_p \leftarrow w_p +$ Average($w_p$.buffer)
18: **end for**
19: **return** $P_{t+1}$.

---

**Algorithm A.3** The **Decode**($C, v_c$) Primitive

---

**Input:** $C$ (the carrier model), $v_c$ (the integer secret of the carrier model), $N_w, N_b, N_s$ (the length of $P_w, P_b, P_s$).
**Output:** $P = P_w \cup P_b \cup P_s$ (the decoded weight pool).
1: $P_w \leftarrow list(), P_b \leftarrow list(), P_s \leftarrow list()$
2: **for** each scalar parameter $w$ in $C$ **do**
3:     **if** $w$ belongs to a {weight, scale, bias} parameter **then**
4:         **switch** (type($w$)) **do**
5:         **case** weight**:**
6:             $P_w$.Append($w$.data)
7:         **case** bias**:**
8:             $P_b$.Append($w$.data)
9:         **case** scale**:**
10:             $P_s$.Append($w$.data)
11:         **end switch**
12:     **else**
13:         CONTINUE
14:     **end if**
15: **end for**
16: $v_w, v_b, v_s \leftarrow v_c \mod N_w, v_c \mod N_b, v_c \mod N_s$
17: Generate a random permutation $\pi_w$ ($\pi_b, \pi_s$) of integers from 0 to $N_w$ ($N_b, N_s$) with seeds $v_w(v_b, v_s)$.
18: $P_w, P_b, P_s \leftarrow \text{Fusion}(P_w, P_b, P_s)$      ▷ Recover the parameter by selecting the non-zero value from each weight pool copy, if needed.
19: Right-shift each parameter group by $v_w, v_b, v_s$.
20: $P_w, P_b, P_s \leftarrow \pi_w^{-1} \circ P_w, \pi_b^{-1} \circ P_b, \pi_s^{-1} \circ P_s$
21: **return** $P = P_w \cup P_b \cup P_s$

---

blocking data transfer to unauthorized endpoint devices (e.g., Wi-Fi, Bluetooth, USB) or to the outside network [2], various access control policies are implemented by AI corporations to manage their owned internal datasets. In the following, we discuss two typical modes named as the *Willing-to-Share* mode and the *Application-then-Authorization* mode informed by the industry partners.

- For start-up corporations which focus on one killer application (e.g., object detection), they prefer to organize the datasets in a more shareable mode. According to our field study, a majority of the groups organize all the datasets (either from public or private sources) in their local distributed file system, which enables fast access to specific datasets according to one's requirement and facilitates swift development of novel DL techniques.

- More established AI corporations prefer to implement a more conservative way of managing ML datasets. According to Google's common security whitepaper [1], each employer should first apply for the authorization before accessing certain data resources, including the access to private datasets. However, considering the ever-evolving paradigms in deep learning, employees with ulterior motives may fabricate reasons such as the requirements of data augmentation [6] or the purpose of multimodal learning [3] to apply for relevant and irrelevant private datasets, which is common in social engineering [4]. Under the umbrella of DLPS, corporations may be less precautious about the unnecessary access to certain private datasets, as DLPS ought to forbid any attempts of transferring the private datasets away from the local network.

However, neither of the above notes can prevent an insider to access the private dataset, especially when he/she is assigned with workloads related to the private dataset or applies to access the private dataset for, e.g., *multimodal learning* (i.e., a learning paradigm which leverages multiple datasets from different domains to enhance the model training [3]).

## C   Omitted Experimental Details

### C.1   Detailed Architecture of Secret Models

We provided the detailed architecture of secret models on CIFAR-10, FaceScrub, and SpeechCommand in Table C.1, C.2, and C.3 respectively.

Table C.1: The detailed architecture of secret generator on CIFAR-10, which is described by convention of PyTorch.

|  |  |
| --- | --- |
| **Generator** | nn.ConvTranspose2d(100, 512, 4, 1, 0, bias=False) |
|  | nn.BatchNorm2d(512) |
|  | nn.ReLU() |
|  | nn.ConvTranspose2d(512, 256, 4, 2, 1, bias=False) |
|  | nn.BatchNorm2d(256) |
|  | nn.ReLU() |
|  | nn.ConvTranspose2d(256, 128, 4, 2, 1, bias=False) |
|  | nn.BatchNorm2d(32) |
|  | nn.ReLU() |
|  | nn.ConvTranspose2d(128, 64, 4, 2, 1, bias=False) |
|  | nn.BatchNorm2d(64) |
|  | nn.ReLU() |
|  | nn.ConvTranspose2d(64, 3, 1, 1, 0, bias=False) |
|  | nn.Tanh() |

### C.2   Other Experimental Setups

For other common hyper-parameter settings in Algorithm A.2, we train our carrier model for 200 epochs using Stochastic Gradient Descent(SGD) with an initial learning rate of 0.1, a weight decay of $5 \times 10^{-4}$, and a momentum of 0.9. We trained our secret models using Adam optimizer with an initial learning rate of $2 \times 10^{-4}$ on CIFAR-10 and SpeechCommand ($1 \times 10^{-3}$ on FaceScrub), and running average coefficients of 0.5, 0.999. For faster convergence, we dynamically adjust the

Table C.2: The detailed architecture of secret generator on FaceScrub, which is described by convention of PyTorch.

| | |
|---|---|
| **Generator** | nn.ConvTranspose2d(100, 512, 7, 1, 0, bias=False) |
| | nn.BatchNorm2d(512) |
| | nn.ReLU() |
| | nn.ConvTranspose2d(512, 256, 4, 2, 1, bias=False) |
| | nn.BatchNorm2d(256) |
| | nn.ReLU() |
| | nn.ConvTranspose2d(256, 128, 4, 2, 1, bias=False) |
| | nn.BatchNorm2d(32) |
| | nn.ReLU() |
| | nn.ConvTranspose2d(128, 64, 4, 4, 0, bias=False) |
| | nn.BatchNorm2d(64) |
| | nn.ReLU() |
| | nn.ConvTranspose2d(64, 3, 4, 2, 1, bias=False) |
| | nn.Tanh() |

Table C.3: The detailed architecture of secret generator on SpeechCommand, which is described by convention of PyTorch.

| | |
|---|---|
| **Generator** | nn.ConvTranspose1d(100, 64, 3, 1, 0, bias=False) |
| | nn.BatchNorm1d(64) |
| | nn.ReLU() |
| | nn.ConvTranspose1d(64, 64, 27, 3, 0, bias=False, dilation=2) |
| | nn.BatchNorm1d(64) |
| | nn.ReLU() |
| | nn.ConvTranspose1d(64, 32, 38, 3, 0, bias=False, dilation=2) |
| | nn.BatchNorm1d(32) |
| | nn.ReLU() |
| | nn.ConvTranspose2d(32, 32, 52, 3, 2, bias=False, dilation=5) |
| | nn.BatchNorm1d(32) |
| | nn.ReLU() |
| | nn.ConvTranspose1d(32, 1, 80, 16, 0, bias=False) |
| | nn.Tanh() |

learning using a cosine annealing schedule with the number of epochs $T_{max}$ of 200. We set the batch size of CIFAR-10, FaceScrub, and SpeechCommand as 128, 64, and 64 respectively.

All the experiments are implemented with Torch [5], which is an open-source software framework for numeric computation and deep learning. All our experiments are conducted on a Linux server running Ubuntu 16.04, one AMD Ryzen Threadripper 2990WX 32-core processor and 2 NVIDIA GTX RTX2080 GPUs.