# OpenReview forum: "House of Cans: Covert Transmission of Internal Datasets via Capacity-Aware Neuron Steganography"
_NeurIPS.cc/2022/Conference — NeurIPS 2022 Accept_

### Official Review · Reviewer_Lxtr · 2022-07-09

**Rating:** 8
**Confidence:** 4
**Soundness:** 3 good
**Presentation:** 3 good
**Contribution:** 4 excellent

**Summary:**

This paper presents a method called Cans for encoding secret datasets into deep neural networks (DNNs) and transmitting them in a openly shared “carrier” DNN. In contrast to existing steganography methods encoding information into least significant bits, the authors encode the secret dataset into a trained publicly shared DNN model such that the public model will predict weights for secret key inputs (known to covert operatives), the weights are used to populate a secret DNN model and the secret DNN model predicts secret dataset for noisy inputs (known to covert operatives). The main advantage of the Cans encoding is that it can covertly transmit over 10000 real-world data samples within a carrier model which has 100× less parameters than the total size of the stolen data, and simultaneously transmit multiple heterogeneous datasets within a single carrier model, under a trivial distortion rate (< 10−5) and with almost no utility loss on the carrier model (< 1%).

**Questions:**

•	Please, fix the typos including the title (Stegnography), “Under the umberalla of DLPS,”
•	How is encoding a secret dataset into a neural network model different from injecting a trojan into a neural network model, for instance, in the TrojAI program – see https://pages.nist.gov/trojai/docs/about.html?
•	Why is the method denoted as “capacity-aware” as opposed to “capacity-limited”? The encoding of secret dataset is either successful or not, but it is not probing the carrier model for its capacity, correct?
•	What is the weight pool restoration algorithm?
o	“We initialize the weight pool with a customized size 5× smaller than the carrier model, which allows us to implement a weight pool restoration algorithm to recover the pruned parameter by selecting the non-zero value from each weight pool copy, i.e., the fusion mechanism. “


**Limitations:**

•	How is the information redundancy built into the Fill, Propagate, Decode algorithms?
o	In reference to the sentence “ Finally, by comparing the performance of the secret model with or without fusion, we conclude that the robustness of Cans largely comes from the information redundancy implemented in our design of the weight pool”


**Strengths And Weaknesses:**

Strength:
•	The authors nicely combine cryptographic application and DNN modeling.  The idea of hiding the secret dataset in a shared model parameters is very interesting.
•	The authors also nicely presented their experimental work with accuracy and robustness evaluations.

Weaknesses:
•	The paper is missing an integration of the main algorithmic steps (Fill, Propagate, Decode) with the overarching flow diagram in Fig 1 which creates a gap in the presentation.

•	The abstract and main text make inconsistent claims about the transmission capacity:
o	Abstract: “.. covertly transmit over 10000 real-world data samples within a carrier model which has 220× less parameters than the total size of the stolen data,”
o	Introduction: “… covertly transmit over 10000  real-world data samples within a carrier model which has 100× less parameters than the total size of the stolen data (§4.1),”

•	Definitions of metrics and illustrations of qualitative results are insufficiently described and included.
o	For example, the equation for a earning objective in section 3.3 should be clearly described.
o	Page 7: define performance difference and hiding capacity in equations.
o	Fig 3 is too small for the information to be conveyed (At the 200% digital magnification of Fig 3, I can see some differences in image qualities).

•	The choices and constructions of a secret key and noisy vectors are insufficiently described i.e., Are the secret keys similar to the public-private keys used in the current cryptography applications? What are the requirements on creating the noisy vectors?

---

> ### Author Response · Authors · 2022-08-01
> **Response to Official Review of Paper4751 by Reviewer Lxtr**
>
>
> Thank you for the encouragement on our work. Below, we reply one by one to the comments.
>
> **W1.** ***The paper is missing an integration of the main algorithmic steps (Fill, Propagate, Decode) with the overarching flow diagram in Fig 1 which creates a gap in the presentation.***
>
> **Re:** We will refine Fig.1 in our paper to integrate the main algorithmic steps in our camera-ready version.
>
> **W2.** ***The abstract and main text make inconsistent claims about the transmission capacity.***
>
> **Re:** We have fixed the typos in the claims of the introductory part in the rebuttal revision. According to the statistics in Section 4, the size of 10000 real-world data samples from FaceScrub is 220x larger than the size of the carrier model.
>
> **W3.** ***Definitions of metrics and illustrations of qualitative results are insufficiently described and included.***
>
> **Re:** We will add more explanation on the learning objective in Section 3.3, define the performance difference and hiding capacity in equations in Section 7, and refined the presentation of Fig.3 in the camera-ready version.
>
> **W4.** ***The choices and constructions of a secret key and noisy vectors are insufficiently described. What are the requirements on creating the noisy vectors?***
>
> **Re:** We will incorporated more discussion on the choices and the constructions of the secret keys in Section 3 of the camera-ready version. The secret keys in our scheme is more similar to the one in the symmetric key cryptography. Concisely, our schemes adopt two groups of secret keys:
>
> (1) The first group contains (C+1) secret integers randomly sampled from [0, |P|) (i.e., |P| is the size of the weight pool, which is at the scale of 10^8 in our experiments, and C is the number of secret tasks). Used in the **Fill** algorithm, the first secret integer determines how the weight pool is filled into the carrier model, while the next C secret integers determine how the weight pool is filled in the secret models
>
> (2) The second group contains C secret random integers sampled from [0, MAX_INT]. The i-th integer severs as the random seed for the i-th secret task. Specifically, a set of noise vectors are generated under the given random seed and are used in each secret task as a one-to-one correspondence with the victim data. In our current implementation, the noise vectors are sampled from a standard Gaussian under the given random seed.
>
> **Q1.** ***Please, fix the typos including the title (Stegnography), “Under the umberalla of DLPS,”***
>
> **Re:** Thank you for the careful reviewing. We have fixed the typos in the rebuttal version.
>
> **Q2. *How is encoding a secret dataset into a neural network model different from injecting a trojan into a neural network model***
>
> **Re:** This is a very interesting question.  From our perspective, the trojan/backdoor attacks on DNN (e.g., BadNet, TrojanNN, Composite Backdoor, etc.) mainly modifies the victim model so that it can triggered by attacker-specified inputs to make targeted misclassification. This is mainly for evasive purposes (e.g., evading a face recognition system). Differently, our attack modifies the carrier model to hold a privacy backdoor, i.e., a secret dataset can be decoded from the normal model via an attacker-specified procedure.
>
> **Q3.** ***Why is the method denoted as “capacity-aware” as opposed to “capacity-limited”? The encoding of secret dataset is either successful or not, but it is not probing the carrier model for its capacity, correct?***
>
> **Re:** We are afraid but our approach indeed exploits the learning capacity of deep neural networks (i.e., both the carrier and the secret models) to hide more samples from the secret datasets. In our scheme (i.e., Section 3.3), the encoding of the secret dataset is converted into a learning objective w.r.t. the weight pool. Therefore, the outcome of the encoding process is not simply successful or not. When the victim dataset is over the learning capacity, the recovered images can still be recognizable despite a certain level of distortion (cf. Fig.4(b)).
>
> **Q4.** ***What is the weight pool restoration algorithm?***
>
> **Re:** When the weight pool is set to be N times smaller than the carrier model, there would be N-redundancy for the weight pool encoded in the carrier model. A general weight pool restoration algorithm works by first extracting the N copies of the weight pool from the carrier model and fuse them into the final weight pool. For example, if the carrier model undergoes pruning, the fusion works by assembling the final weight pool from the non-zero (i.e., unpruned) values from each copy. In this way, the final weight pool can be almost recovered despite each copy may have certain values to be pruned.

---

### Official Review · Reviewer_5E9b · 2022-07-13

**Rating:** 5
**Confidence:** 3
**Soundness:** 3 good
**Presentation:** 3 good
**Contribution:** 3 good

**Summary:**

This paper presents a new steganography model to recover the images from noise messages. It has two key contributions. First, it proposes a weight pool such that the model is assembled from the weights in this pool with the secret key v. People are hard to get the correct model with the key. Second, they design a trainable pipeline to co-train the generated model and classification model with the weight pool. The authors evaluate the model on three different datasets with SSIM, MSE, and Performance Difference metrics.

**Questions:**

1. How many secret keys did the authors use when training the model? I think if we use too few secret keys, the stealer might be easy to recover the model by credential stuffing.
2. How long will it take to train the model? Does training with the weight pool make the training process very slow?


**Ethics Review Area:**

["Privacy and Security (e.g., consent)"]

**Limitations:**

This paper proposes an image steganography method. One possible negative societal impact is people might use it to steal secret information.

**Strengths And Weaknesses:**

Strengths
1. The key idea sounds novel to me. The generation model is assembled from a weight pool by using the private key. Therefore, if stealers do not know the key, it is hard for them to get the correct generation model.
2. The authors proposed a novel training pipeline to train the generation model with the weight pool.
3. The results look promising.
4. The authors provide the details to implement and train the model in the supplementary.

Weakness:
1. There is no related work section. Please include related image steganography models and generative models in the related work.
2. I have a concern that this paper does not include any baselines. It is hard for me to get a sense of how good the proposed model is. [1] [2] use a similar metric to evaluate the steganography performance. Please consider comparing the model with them.
3. The algorithms in the supplement are a little hard to understand. Please consider adding some explanation. It is the key concept of the paper. Please also consider moving them to the main paper.

[1] Zhang, Kevin Alex, et al. "SteganoGAN: High capacity image steganography with GANs." arXiv preprint arXiv:1901.03892 (2019).
[2] Kishore, Varsha, et al. "Fixed Neural Network Steganography: Train the images, not the network." International Conference on Learning Representations. 2021.

---

> ### Author Response · Authors · 2022-08-01
> **Response to Official Review of Paper4751 by Reviewer 5E9b**
>
> Thank you for the encouragement on our work. Below, we reply one by one to the comments.
>
> **W1.** ***There is no related work section. Please include related image steganography models and generative models in the related work.***
>
> **Re:** We have incorporated a brief review on the related image steganography models and generative models in the "*Data Hiding in the Deep Learning Era*" part of Section 2 of our rebuttal revision.
>
> **W2. *I have a concern that this paper does not include any baselines. It is hard for me to get a sense of how good the proposed model is. [1] [2] use a similar metric to evaluate the steganography performance. Please consider comparing the model with them.***
>
> **Re:** In Section 4, we did incorporate three baseline approaches which also attempt to hide data in a deep neural network via conventional steganography approaches (Ref. [39] of our paper). As shown in Fig.3, our capacity-aware approach outperforms all of the baselines in the hiding capacity. For example, in Fig.3(a), the baseline approaches could not be executed to hide over 4096 images from CIFAR-10, while our approach still works and the recovered images remain close to the original ones in both SSIM (Fig.3(a)) and the perceptual quality (Fig.4(b)). To the best of our knowledge, the covered baselines are the only ones which use the deep neural networks as the carrier medium to hide information. We are afraid but [1][2] mainly exploits deep learning techniques (e.g., GAN or encoder-decoder) towards image steganography, i.e., taking an image as the carrier medium and hiding secret bit strings in the image. Therefore, their task is different from ours and could not serve as a baseline. Nevertheless, we have incorporated the discussion on such works in Section 2 of our rebuttal revision.
>
> **W3. *The algorithms in the supplement are a little hard to understand. Please consider adding some explanation. It is the key concept of the paper. Please also consider moving them to the main paper.***
>
> **Re:** We will incorporate the explanation on the algorithms in our main text also in the supplements to make it self-contained in the camera-ready version. Besides, we will put a simplified version of the algorithms in the supplements in our main text.
>
> **Q1.** ***How many secret keys did the authors use when training the model? I think if we use too few secret keys, the stealer might be easy to recover the model by credential stuffing.***
>
> **Re:** We also agree that using few secret keys would expose the encoded training data under the risk of being decoded by irrelevant users. Therefore, our schemes adopt two groups of secret keys, which are rather complicated for credential guessing:
>
> **(1)** The first group contains (C+1) secret integers randomly sampled from [0, |P|) (i.e., |P| is the size of the weight pool, which is at the scale of 10^8 in our experiments, and C is the number of secret tasks). Used in the **Fill** algorithm, the first secret integer determines how the weight pool is filled into the carrier model, while the next C secret integers determine how the weight pool is filled in the secret models
>
> **(2)** The second group contains C secret random integers sampled from [0, MAX_INT]. The i-th integer severs as the random seed for the i-th secret task. Specifically, a set of noise vectors are generated under the given random seed and are used in each secret task as a one-to-one correspondence with the victim data.
>
> During the decoding phase, the adversary first uses the first set of secret integers to decode the weight pool from the carrier model and constructs the C secret models with the weight pool. Then, he/she uses the second set of secret integers to generate the same set of noise vectors used in the encoding phase and feeds them into the secret models to dump the victim data.
>
> **Q2.** ***How long will it take to train the model? Does training with the weight pool make the training process very slow?***
>
> **Re:** To compare the overhead of introducing weight pool in training, we conduct a demonstrative experiment in the rebuttal phase. Specifically, we consider three comparison groups: (i) Normal Training (i.e., w/o. Weight Pool, w/o. secret task), (ii) Normal Training with Weight Pool (i.e., w/. Weight Pool, w/o. secret task), (iii) Training with both weight pool and one secret task (i.e., w/. Weight Pool, w/. secret task). The following table compares the time of training 100 iterations under the three configurations (with 10 repetitive tests in the same computing environments). As is shown, the introduction of the weight pool and the joint training with secret tasks won’t make the training process very slow.
>
> |  | (i) w/o. weight pool, w/o. secret task  | (ii) w/. weight pool, w/o. secret task  | (iii) w/o. weight pool, w/o. secret task  |
> | --- | --- | --- | --- |
> | Time Cost (sec./100 iteration) | 9.80 ( $\pm$ 0.09) | 12.57  ( $\pm$ 0.04) | 15.08 ( $\pm$ 0.04) |

---

### Official Review · Reviewer_kzME · 2022-07-18

**Rating:** 5
**Confidence:** 2
**Soundness:** 3 good
**Presentation:** 3 good
**Contribution:** 3 good

**Summary:**

This paper studies how to use models to hide data.

**Questions:**

Since I am not very familiar with this topic, I am willing to adjust my score based on the questions from other reviewers and rebuttal from the authors.

**Ethics Review Area:**

["I don’t know"]

**Limitations:**

N.A.

**Strengths And Weaknesses:**

Strengthes: I checked the performance, which seems to be very impressive and is the main reason I argue for acceptance. The paper is also well written and organized. Reasoanble analysis is provided to justify the reported results. I do not see a clear weakness of this paper. Some papers might be worth a citation, such as UDH: universal deep hiding from NeurIPS2020. To my knowledge, it is not a common practice to publish this topic at NeurIPS. Maybe it is worth a discussion.

---

> ### Author Response · Authors · 2022-08-01
> **Response to Official Review of Paper4751 by Reviewer kzME**
>
> Thank you for the encouragement on our work. We have incorporated the discussion on image steganography via deep learning techniques (including [1] which kindly pointed out by the reviewer) in the "*Data Hiding in the Deep Learning Era*" part of Section 2 of our rebuttal revision.
>
> [1] UDH: Universal Deep Hiding for Steganography, Watermarking, and Light Field Messaging, NeurIPS'20

---

### Meta-Review · Area_Chair_TKKL · 2022-08-27

**Recommendation:** Accept
**Confidence:** Less certain

**Metareview:**

The reviewers are generally positive, with Reviewer Lxtr being mostly enthusiastic about the work. Both Reviewer Lxtr and 5E9b believe the work of encrypting and transmitting secrete data with DNN is novel and the work is well executed with thorough experimental results and technical details. The reviewers raised several questions about the related work and baselines, as well as algorithmic clarity. The authors should further address these points by incorporating additional discussions and results from the rebuttal in the revision.

**Award:**

No

---

### Decision · Program_Chairs · 2022-09-14

Accept